# Dual HLA B*42 and B*81-reactive T cell receptors recognize more diverse HIV-1 Gag escape variants

Funsho Ogunshola[1,2], Gursev Anmole[3], Rachel L. Miller[4], Emily Goering[5], Thandeka Nkosi[1,2], Daniel Muema[1], Jaclyn Mann[2], Nasreen Ismail[2], Denis Chopera[1], Thumbi Ndung'u[1,2,5,6], Mark A. Brockman [3,4,7] & Zaza M Ndhlovu [1,2,5]

Some closely related human leukocyte antigen (HLA) alleles are associated with variable clinical outcomes following HIV-1 infection despite presenting the same viral epitopes. Mechanisms underlying these differences remain unclear but may be due to intrinsic characteristics of the HLA alleles or responding T cell repertoires. Here we examine CD8$^+$ T cell responses against the immunodominant HIV-1 Gag epitope TL9 (TPQDLNTML$_{180-188}$) in the context of the protective allele B*81:01 and the less protective allele B*42:01. We observe a population of dual-reactive T cells that recognize TL9 presented by both B*81:01 and B*42:01 in individuals lacking one allele. The presence of dual-reactive T cells is associated with lower plasma viremia, suggesting a clinical benefit. In B*42:01 expressing individuals, the dual-reactive phenotype defines public T cell receptor (TCR) clones that recognize a wider range of TL9 escape variants, consistent with enhanced control of viral infection through containment of HIV-1 sequence adaptation.

[1] Africa Health Research Institute, University of KwaZulu-Natal, Durban, South Africa. [2] HIV Pathogenesis Programme, Doris Duke Medical Research Institute, University of KwaZulu-Natal, Durban, South Africa. [3] Department of Molecular Biology and Biochemistry, Simon Fraser University, Burnaby, BC V5A 1S6, Canada. [4] Faculty of Health Sciences, Simon Fraser University, Burnaby, BC V5A 1S6, Canada. [5] Ragon Institute of MGH, MIT, and Harvard, Cambridge, MA 02139, USA. [6] Max Planck Institute for Infection Biology, Berlin, Germany. [7] British Columbia Centre for Excellence in HIV/AIDS, Vancouver, BC V6Z 1Y6, Canada. These authors contributed equally: Funsho Ogunshola, Gursev Anmole. These authors jointly supervised this work: Mark A. Brockman, Zaza Ndhlovu. Correspondence and requests for materials should be addressed to M.A.B. (email: mark_brockman@sfu.ca) or to Z.M.N. (email: zndhlovu@mgh.harvard.edu)

The rate of clinical progression following human immunodeficiency virus type 1 (HIV-1) infection is variable, with rare individuals maintaining plasma viral loads below 50 RNA copies mL$^{-1}$ in the absence of therapy[1,2]. Host and viral mechanisms associated with relative control of infection indicate that the ability of HIV-1 to adapt to a new host is a critical determinant of pathogenesis[3,4]. Multiple lines of evidence support the central role of CD8$^+$ T cells in this process[5–7]. Expression of certain class I human leukocyte antigen (HLA) alleles, particularly at the HLA-B locus[8,9], is associated with lower plasma viral loads, higher CD4$^+$ T cell counts and delayed onset of AIDS[10,11]. Interaction between CD8$^+$ T cells and viral peptide epitopes presented on HLA determines breadth and other characteristics of the antiviral response[12,13], while rapid development of viral mutations in targeted epitopes facilitates evasion from host immunity[3,14,15]. CD8$^+$ T cells that target epitopes derived from p24 Gag are associated with better control[16,17], likely due to their relative immunodominance and greater fitness constraints on this major viral structural protein[15,18–20].

Recognition of a peptide/HLA (pHLA) ligand by a CD8$^+$ T cell is determined by the sequence and functional characteristics of its T cell receptor (TCR)[21,22]. The exceptional diversity of the TCR repertoire, generated by somatic recombination of variable (V), diversity (D), and joining (J) gene segments, junctional modifications, and differential pairing of α and β chains, has profound implications for immune coverage[23]. In addition to defining antigen specificity, TCR affinity for pHLA can dictate the strength of intracellular signaling events that modulate T cell effector functions, including cytotoxicity and proliferative capacity[24]. Characteristics of TCR clonotypes that contribute most effectively to CD8$^+$ T cell-mediated control of HIV-1 infection are largely unknown, since data linking individual TCR sequences with measures of antiviral function remains limited. In previous studies of p24 Gag epitopes TW10 (TSTLQEQIGW$_{240–249}$) and KK10 (KRWIILGLNK$_{263–272}$), presented on protective HLA alleles B*57:01 and B*27:05, respectively, CD8$^+$ T cell clones displaying higher functional avidity or greater ability to cross-recognize epitope variants were shown to have enhanced antiviral activity[25–28]. In the case of B*27-KK10, public TCR clonotypes, defined as having identical (or nearly identical) TCR β sequences in the antigen-specific repertoire of at least two unrelated individuals[22,29], displaying high avidity against the consensus epitope were also associated with a more effective T cell response[28,30].

Following infection with HIV-1 subtype C strains that are prevalent in sub-Saharan Africa, expression of HLA allele B*81:01 is associated with improved clinical outcomes[9,31], while the genetically-related allele B*42:01 is less protective[16,31–36]. Both alleles belong to the HLA B7 supertype[37,38] and present similar viral peptides, including the immunodominant p24 Gag epitope TL9 (TPQDLNTML$_{180–188}$)[34,39–42]. The magnitude of the TL9 response has been associated with lower plasma viremia and improved clinical outcome in the case of B*81:01[43]. TL9 is located on helix 3 of the p24 protein, which is critical to form the mature viral capsid. Circulating subtype C strains display >99% sequence identity at all TL9 residues except positions 3 (88.5%) and 7 (93.5%) (HIV Databases; www.hiv.lanl.gov). Positions 3 and 7 are the principal sites for viral escape from CD8$^+$ T cell pressure[3,31,40,41]; however, mutations at these residues also impair fitness[44], indicating that HIV-1 adaptation at TL9 must balance these counteracting pressures. Structural studies indicate that the TL9 residues exposed to T cells differ in its bound conformations with B*81:01 and B*42:01[45], and some evidence suggests that enhanced antiviral T cell function is related to distinct TCR sequences elicited in the context of B*81:01[40]. These observations are consistent with delayed viral escape in B*81:01

expressing individuals compared to B*42:01 expressing individuals[31] and selection of TL9 escape mutations by B*81:01 that tend to be more difficult to compensate for[44]. An improved understanding of clonotypic differences among CD8$^+$ T cells responding to TL9 could highlight features that contribute to HIV-1 control in the context of B*81:01 and B*42:01.

Here we investigate the mechanisms associated with immune-mediated control of HIV-1 subtype C infection by examining the CD8$^+$ T cell response to the immunodominant p24 Gag epitope TL9 in virus-infected individuals expressing HLA B*81:01 or B*42:01 alleles. We identify a subset of T cells that recognize TL9 epitope presented on both B*81:01 and B*42:01 alleles, despite individuals lacking one allele. The presence of a dual-reactive T cell population is associated with lower plasma viral loads after controlling for differences in HLA expression. Notably, the dual-reactive population in B*42:01 expressing individuals is dominated by several public TCR clonotypes that encoded TRBV12-3. In contrast, while mono- and dual-reactive populations in B*81:01 expressing individuals are enriched for TRBV12-3 usage, no public clonotypes are observed. Comprehensive in vitro functional analyses of selected TCR clones demonstrated that B*81:01-derived clones and public dual-reactive B*42:01-derived clones display greater ability to cross-recognize HIV-1 Gag TL9 escape pathways compare to mono-reactive TCR clones isolated from B*42:01 expressing individuals. These results illustrate a use of HLA tetramers and in vitro functional assays to identify and characterize TCR clonotypes that display enhanced ability to recognize a rapidly evolving HIV-1 infection.

## Results

**Characterizing CD8$^+$ T cell responses in study participants.** Population-level studies have demonstrated that HLA-B*81:01 is associated with better control of HIV-1 subtype C infection than the closely related allele B*42:01[3,9]; however, mechanisms to explain this remain unclear. To examine this, we recruited 21 treatment-naive HIV-infected individuals expressing B*81:01 ($n = 9$), B*42:01 ($n = 11$), or both alleles ($n = 1$) from Durban, South Africa. Individuals co-expressing other protective class I HLA alleles (namely B*57:03, B*58:01, and B*39:01) were excluded from this study. The clinical characteristics and class I HLA genotypes of participants are shown in Table 1 and 2, respectively. Consistent with prior reports[40,46,47], we observed that untreated B*81:01 expressing individuals displayed lower plasma viral loads (median 3.38 log$_{10}$ RNA copies ml$^{-1}$ [IQR 2.36–3.99]) compared to untreated B*42:01 expressing individuals (4.15 log$_{10}$ RNA copies ml$^{-1}$ [IQR 3.40–4.84]) ($p = 0.03$, Mann–Whitney U-test) (Fig. 1a). The difference in CD4 counts between groups was not statistically significant (median 625 cells µl$^{-1}$ in B*81:01 vs. 555 cells µl$^{-1}$ in B*42:01; $p = 0.14$, Mann–Whitney U-test). While the individual who co-expressed HLA-B*81:01 and B*42:01 alleles was not included in our analysis of clinical correlations, this participant displayed the lowest plasma viral load (2.11 log$_{10}$ RNA copies ml$^{-1}$) and highest CD4 count (1002 cells µl$^{-1}$).

Immune targeting of dominant CD8$^+$ T cell epitopes contributes to long-term suppression of HIV-1 viremia[48–50]. The p24 Gag-derived epitope TL9 is immunodominant in both B*81:01 and B*42:01 expressing individuals[39,40], and the magnitude of the TL9 response has been associated with improved clinical outcome in the context of B*81:01[43]. To characterize the TL9 response in our cohort, we quantified antigen-specific CD8$^+$ T cells using B*81:01 and B*42:01 tetramers. We observed no difference in the frequency of tetramer$^+$ CD8$^+$ T cells between individuals expressing B*81:01 (median 2.08%) compared to B*42:01 (1.14%) ($p = 0.50$; Student's T test) (Fig. 1b). Notably, intra-patient comparison of

**Table 1 Demographic and clinical characteristics of the study participants**

| HLA | B*81:01/B*42:01 | B*81:01 | B*42:01 | (B*81:01 vs. B*42:01) P value |
|---|---|---|---|---|
| N | 1 | 9 | 11 | N/A |
| Female n (%) | 1 (100%) | 7 (77.8%) | 10 (90.9%) | 0.57[b] |
| Age (yr) | 24 | 22.5 (22.25–28.5)[a] | 22 (20.5–28.5)[a] | 0.75[c] |
| CD4 counts, cells/mm$^3$ | 1002 | 625 (495–802)[a] | 588 (479–673)[a] | 0.14[c] |
| Viral load, log$_{10}$ | 2.11 | 3.38 (2.36–3.99)[a] | 4.15 (3.40–4.84)[a] | 0.03[c] |

*N/A* not applicable
All values in parenthesis were expressed as interquartile range. Participants with other protective alleles present in the cohort of study were excluded in the study
Excluded alleles: HLA-B*57:03; B*58:01; B*39:01
[a]Values expressed as median (interquartile range)
[b]Statistical test used: Fisher's exact test
[c]Statistical test used: Mann–Whitney test

**Table 2 Detail class I HLA profiles of the study participants**

| Participants | HLA-A | HLA-B | HLA-C |
|---|---|---|---|
| PT1 | 33:03/34:02 | 53:01/81:01 | 04:01/04:01 |
| PT2 | 02:00/34:00 | 14:01/81:01 | 08:00/08:00 |
| PT3 | 01:01/30:01 | 81:01/81:01 | 04:01/18:00 |
| PT4 | 23:01/29:02 | 53:01/42:01 | 03:04/17:00 |
| PT5 | 30:01/34:02 | 35:01/42:01 | 02:10/17:01 |
| PT6 | 43:01/74:01 | 57:01/81:01 | 04:01/07:01 |
| PT7 | 01:01/29:11 | 13:02/81:01 | 06:02/18:01 |
| PT8 | 01:01/29:02 | 45:01/81:01 | 06:02/18:01 |
| PT9 | 23:01/68:02 | 14:02/81:01 | 08:02/18:00 |
| PT10 | 02:05/33:01 | 42:01/58:02 | 07:01/17:01 |
| PT11 | 29:02/29:02 | 42:01/45:01 | 06:02/17:01 |
| PT12 | 26:01/30:02 | 15:18/42:01 | 17:01/18:00 |
| PT13 | 30:01/32:01 | 42:01/58:02 | 06:02/17:01 |
| PT14 | 02:01/30:01 | 42:01/45:07 | 16:01/17:01 |
| PT15 | 01:01/74:01 | 35:01/81:01 | 04:01/18:01 |
| PT16 | 02:05/29:02 | 42:01/45:07 | 16:01/17:01 |
| PT17 | 29:01/30:01 | 15:22/42:01 | 04:01/17:01 |
| PT18 | 30:01/68:02 | 14:02/42:01 | 08:02/17:01 |
| PT19 | 23:01/30:01 | 42:01/57:02 | 07:01/17:00 |
| PT20 | 02:00/34:00 | 14:01/81:01 | 08:00/08:00 |
| PT21 | 30:01/68:01 | 42:01/81:01 | 04:01/17:01 |

responses in either B*81:01 or B*42:01 participants showed that TL9 was the most dominant response compared to other responses ($p = <0.0001$, Student's T test) by both tetramer staining and ELISPOT (Supplementary Fig. 1A–C and Supplementary Table 1). These data are consistent with previous studies[40,45]. The proportion of TL9-specific CD8$^+$ T cells expressing IFN-γ following peptide stimulation was also not significantly different between individuals expressing B*81:01 (median 47%) and B*42:01 (27%) ($p = 0.09$, Student's T test) (Fig. 1c); however, the observed trend in favor of B*81:01 participants is consistent with prior work describing moderately higher TL9-specific IFN-γ secretion and higher functional avidity in the context of B*81:01[40].

**Dual HLA reactivity is associated with lower viral load**. To investigate if there were any qualitative differences in TL9-specific CD8$^+$ T cells restricted by these two HLA alleles, we first made a direct comparison between antigen-specific T cells in the individual who co-expressed B*81:01 and B*42:01. Intriguingly, when we double-stained cells from this individual with both HLA tetramers, we observed a dominant T cell subset that was labeled using the B*81:01-TL9 tetramer as well as a secondary subset that was labeled by both B*81:01-TL9 and B*42:01-TL9 tetramers, which we will refer to as the dual-reactive population (Fig. 1d).

To explore whether the dual-reactive T cell population was unique to this individual, we re-examined all study participants

using both class I HLA tetramers. We observed dual-reactive TL9 responses in the majority of participants, indicating that a subset of CD8$^+$ T cells elicited in the context of both B*81:01 and B*42:01 could cross-recognize TL9 bound to the other class I HLA allele, even when it was not expressed by the host. Representative results for two individuals are also shown in Fig. 1d. The dual-reactive population was seen more frequently in individuals expressing B*81:01 (7 of 9; 78%) compared to B*42:01 (5 of 11; 46%) (Fig. 1e), but this difference was not statistically significant ($p = 0.19$, Student's T test). While CD8$^+$ T cell promiscuity is frequently observed toward peptide variants presented on the same HLA allele, we know of only one prior report that described CD8$^+$ T cell cross-reactivity to the same peptide presented on two different class I HLA alleles[51]. In a multivariable linear regression model, we identified dual-reactivity, but not HLA, as a significant independent predictor of lower plasma viral load in our participants ($p = 0.02$) (Fig. 1f), suggesting that this T cell phenotype is associated with a clinical benefit. We therefore hypothesized that features associated with dual-reactive CD8$^+$ T cells could provide insight into mechanisms of HIV-1 control.

**Constrained Vβ genes in B*42-derived dual-reactive TCR**. The ability of dual-reactive CD8$^+$ T cells to recognize TL9 bound to different, albeit related, class I HLA alleles suggested that they harbored distinct characteristics. Since individual TCR clonotypes have been associated with improved control of HIV-1[26–28,52–55], we analyzed the TCR repertoire found in mono- and dual-reactive TL9-specific T cells. First, we investigated TCR β expression using flow cytometry by co-staining PBMC with B*81:01- and B*42:01-TL9 tetramers plus a cocktail of Vβ-specific antibodies. Representative results for one B*42:01 expressing individual are shown in Fig. 2a. Consistent with prior studies that described a high frequency of *TRBV12-3* gene usage among TL9-specific T cells[40,41], we observed that both mono- and dual-reactive T cells from B*81:01 expressing individuals were highly enriched for Vβ 12-3/12-4 (Fig. 2b). In contrast, while mono-reactive T cells from B*42:01 individuals expressed multiple Vβ families, the dual-reactive T cells from these individuals were highly enriched for Vβ 12-3/12-4 (Fig. 2c). These results suggested that TCR clonotypes expressed by dual-reactive CD8$^+$ T cells elicited in the context of HLA B*42:01 shared distinct features with T cells that dominated TL9 responses elicited by the more protective B*81:01 allele. To confirm these observations, we sorted mono- and dual-reactive T cells using FACS and generated separate TL9-specific cell lines. Similar Vβ staining profiles were observed following ex vivo expansion (Supplementary Fig. 2), confirming that dual-reactive T cells were a bona fide population and not an artifact of tetramer staining.

To gain additional molecular insight into the TCR clonotypes present within each TL9-specific T cell population, we sequenced

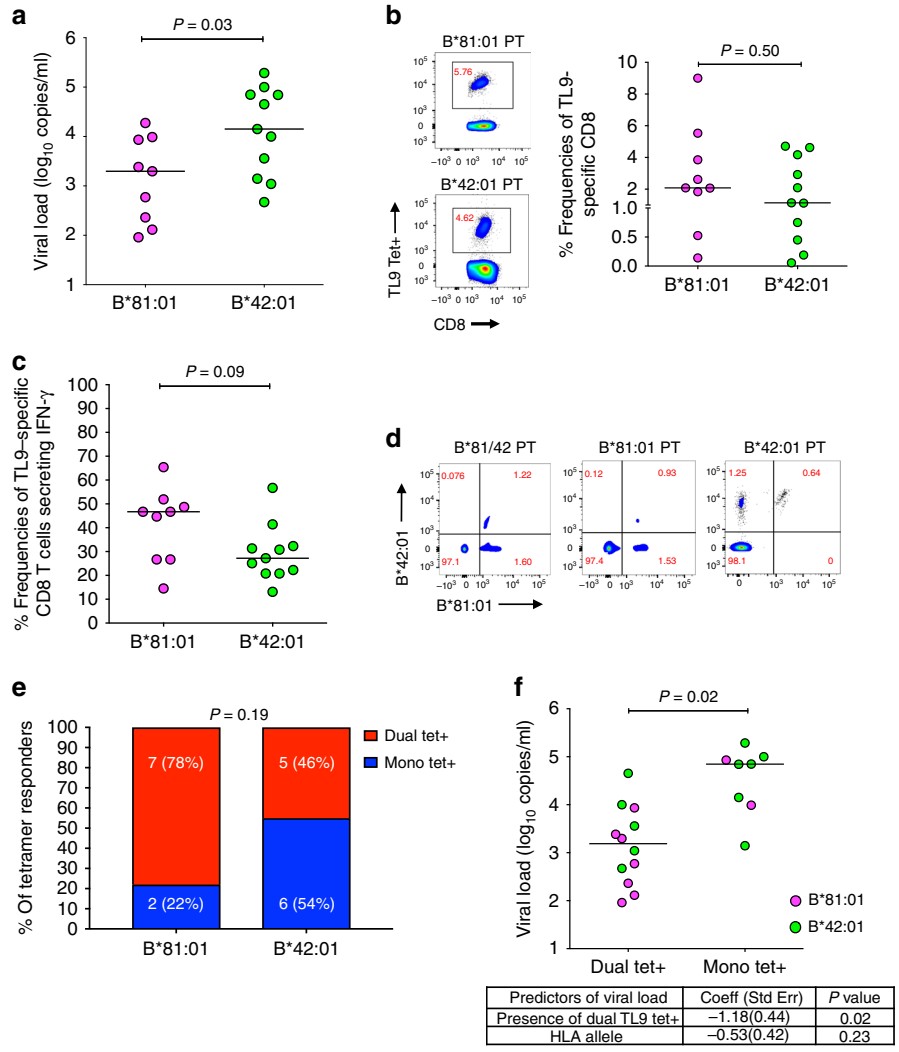

**Fig. 1** A dual TL9 tetramer+ response is associated with lower plasma viral load. A comparative analysis indicated lower plasma viral loads ($\log_{10}$) among participants expressing B*81:01 compared to participants expressing B*42:01 ($p = 0.03$, Mann–Whitney U-test test) (**a**). Representative flow plots display TL9 tetramer responses observed in one B*81:01 expressing individual (top) and one B*42:01 expressing individual (bottom). A comparative analysis of TL9 tetramer+ frequencies observed no difference between participants expressing B*81:01 compared to those expressing B*42:01 ($p = 0.5$; Mann–Whitney U-test) (**b**). A comparative analysis of IFN-γ secretion following stimulation with TL9 peptide indicated a trend toward higher activity among individuals expressing B*81:01 versus B*42:01 ($p = 0.09$, Mann–Whitney U-test) (**c**). Representative flow plots display the dual TL9 tetramer-reactive T cell population in B*81/42:01 expressing participant, one B*81:01 expressing participant and one B*42:01 expressing participant (**d**). A higher proportion of B*81:01 expressing participants displayed dual-tetramer reactivity ($p = 0.19$, Chi-square test) (**e**). Multivariable linear regression analyses that included HLA allele and presence of dual-tetramer reactive T cells as independent variables indicated that dual-reactivity ($p = 0.02$) but not HLA ($p = 0.23$) was a significant determinant of plasma viral load (**f**)

the TCR β gene repertoire in single tetramer-labeled T cells isolated by FACS from three B*81:01 and three B*42:01 expressing individuals who displayed mono- and dual-reactive responses. Consistent with antibody staining results, Vβ gene usage for mono- and dual-reactive B*81:01-derived populations, as well as dual-reactive B*42:01-derived populations, was highly restricted to *TRBV12-3/12-4* (Fig. 3a, b). In contrast, while the mono-reactive population in one B*42:01 expressing individual (participant 11) was comprised largely of T cells encoding *TRBV12-3/12-4*, the primary Vβ gene present in the other two individuals (participants 13 and 17) was *TRBV7-9* (Fig. 3b). Notably, we observed that the dual-reactive population in all three B*42:01 expressing individuals was dominated by four public Vβ sequences (highlighted CDR3 regions in Fig. 3b) that were never observed in B*81:01-derived TCR sequences. Enrichment of *TRBV12-3/12-4* usage by TL9-specific TCR in the context

of B*81:01 as well as the public dual-reactive TCR in B*42:01 expressing individuals suggested that features of these TCR clonotypes contribute to control of HIV-1 infection.

**Isolation and validation of TL9-specific TCR clones**. To provide a more complete understanding of mono- and dual-reactive CD8 + T cell phenotypes, we identified the paired TCR α gene from eight dominant TCR clones representing the mono- and dual-reactive populations from B*81:01 and B*42:01 expressing individuals (Fig. 4a) and directly assessed TCR function using a previously described in vitro reporter T cell assay[56]. Briefly, full-length TCR α/β genes were reconstructed and transiently expressed in Jurkat T cells. TCR-mediated NFAT signaling was quantified by luminescence following co-culture with HLA-expressing target cells presenting the TL9 epitope. Since methods

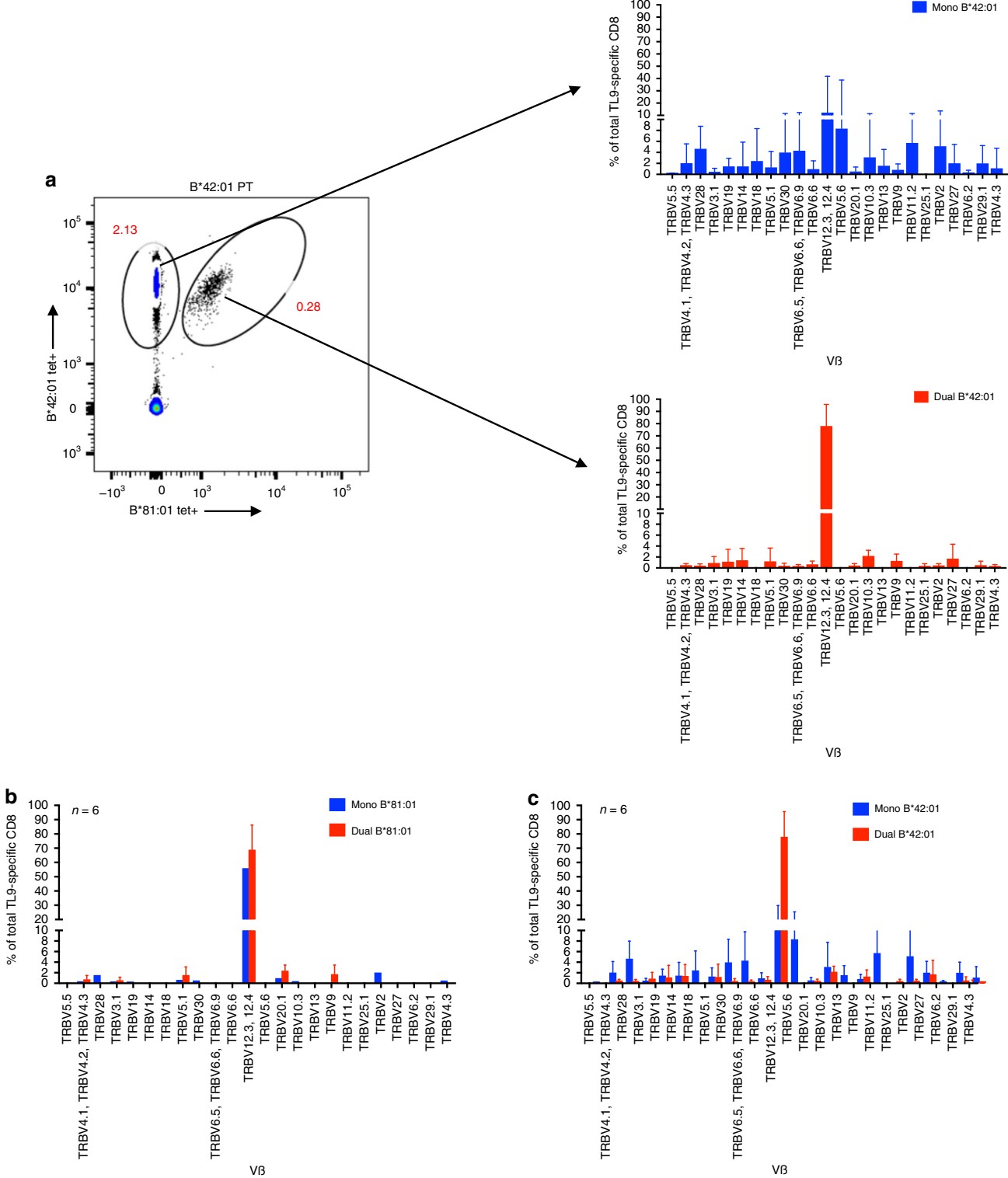

**Fig. 2** Enrichment of TCR Vβ 12–3/12-4 in dual-reactive T cells. A representative flow plot for one B*42:01 expressing individual displays mono- and dual-TL9 tetramer reactive T cell populations and linked TCR Vβ expression profiles based on antibody staining (**a**). Aggregate results for TCR Vβ usage are shown for mono- (blue) and dual-reactive (red) T cells from six B*81:01 expressing individuals (**b**) or six B*42:01 expressing individuals (**c**)

used for TCR staining and sequencing could not distinguish between *TRBV12-3* and *TRBV12-4*, which differ by two amino acids in the CDR1, TCR β genes were synthesized encoding both alleles; however, only *TRBV12-3* constructs were functional (Supplementary Fig. 3). TCR clones displayed dose-dependent

responses to consensus TL9 over a range of peptide concentrations (5 nM–20 μM) (Supplementary Fig. 4), indicating that the reporter assay was sensitive and specific. Furthermore, reconstructed TCR maintained mono- or dual-reactivity against TL9 peptide-pulsed (Fig. 4b) and HIV-infected (Fig. 4c) target cells

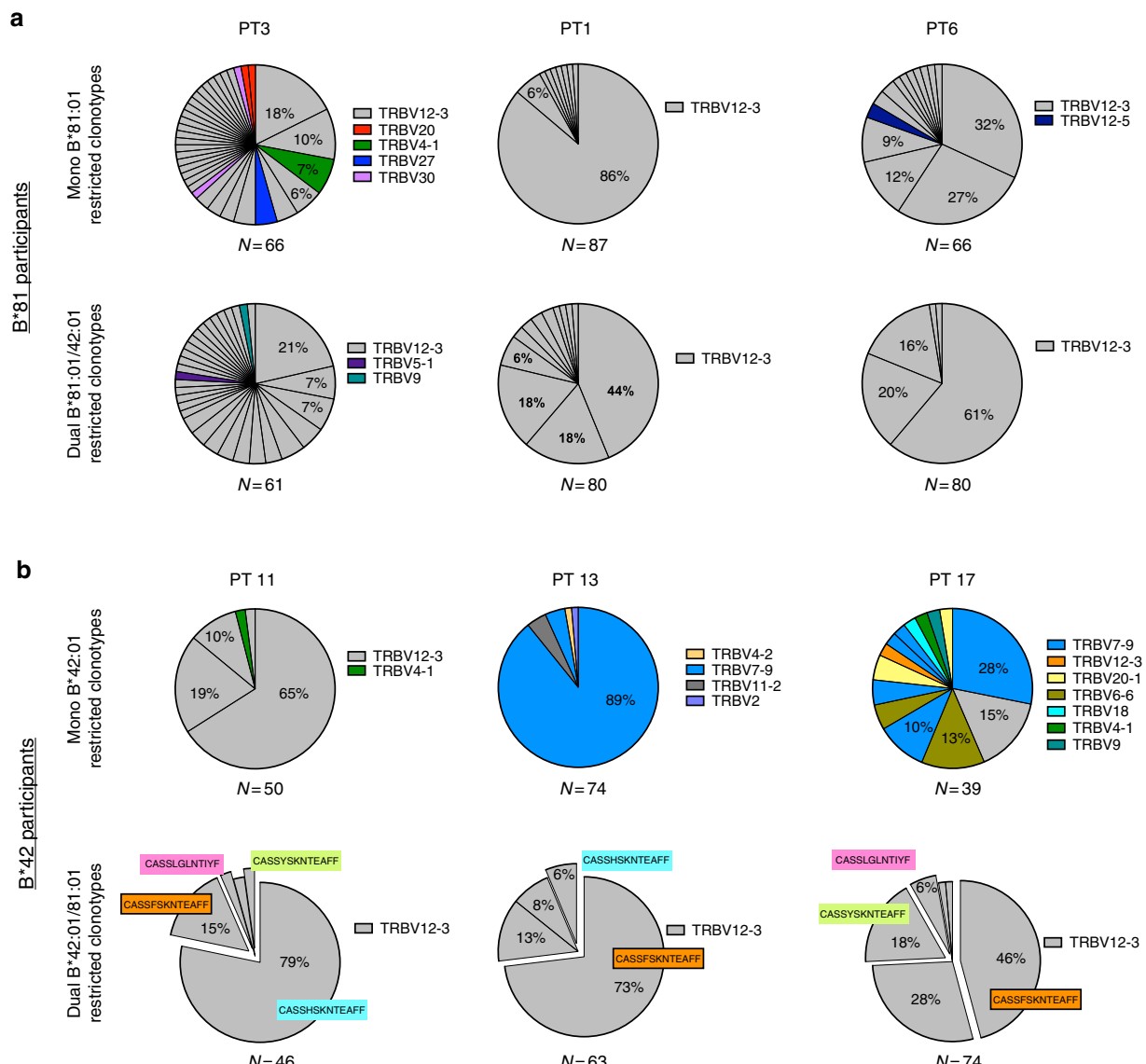

**Fig. 3** Molecular analysis of TCR β clonotypes in mono- and dual-reactive T cells. TCR β sequencing was performed on single FACS-sorted mono- and dual-TL9 tetramer reactive T cells from three B*81:01 expressing participants (**a**) and three B*42:01 expressing participants (**b**). *TRBV* and CDR3 sequences were determined using the IMGT V-quest tool (www.imgt.org). The total number of sequences collected per population is indicated under each pie chart. Unique TCR β clones are displayed as wedges in the pie chart. The size of the wedge indicates the frequency of each sequence within the population and the color represents *TRBV* usage. TCR β sequences in mono- and dual-reactive populations from B*81:01 expressing individuals were highly enriched for *TRBV12-3/12-4* usage (indicated in gray); however, no public sequences were observed among these individuals. Mono-reactive populations from B*42:01 expressing individuals encoded diverse *TRBV* and also lacked public sequences. In contrast, dual-reactive populations from B*42:01 expressing individuals were enriched for *TRBV12-3/12-4* usage (gray), and these sequences were comprised predominantly of four identical (public) TCR β clones (highlighted by colored boxes). Notably, these public clones were distinct from any TCR observed in B*81:01 individuals

expressing B*81:01 or B*42:01, confirming that dual-reactive T cells were a distinct population in both B*81:01 and B*42:01 expressing individuals, and that phenotypic differences in pHLA specificity were due to TCR sequence.

**Analyses of TL9 variant recognition by TCR clones.** The ability of TCR to cross-recognize epitope variants is associated with enhanced antiviral activity of CD8⁺ T cells[26,27,57]. If indeed the dual-reactive population contributes to control of HIV-1, we hypothesized that it should be able to respond to a variety of TL9 variants. To explore this, we assessed the ability of each reconstructed TCR to respond to a panel of 180 peptides representing

TL9 and all possible single amino acid TL9 variants. These results are displayed as heat maps in Fig. 5 and also provided in the Supplementary Data file. Collectively, the eight TCR clones recognized 114 (of 171, 67%) TL9 variants at a normalized luminescence value of 0.1 or greater (which was ~10-fold above negative control wells). In addition to consensus TL9, individual B*81:01-derived clones responded to 67 (11A10, mono-reactive), 53 (12A11, dual), and 94 (18A2, dual) variant peptides, while B*42:01-derived clones responded to 48 (7A10, mono), 46 (13A10, mono), 54 (14A4, public dual), 34 (16A11, public dual), and 34 (14D7, public dual) variant peptides. No correlation was observed between the total breadth of TL9 variant recognition and dual-reactivity, suggesting that qualitative features of TCR

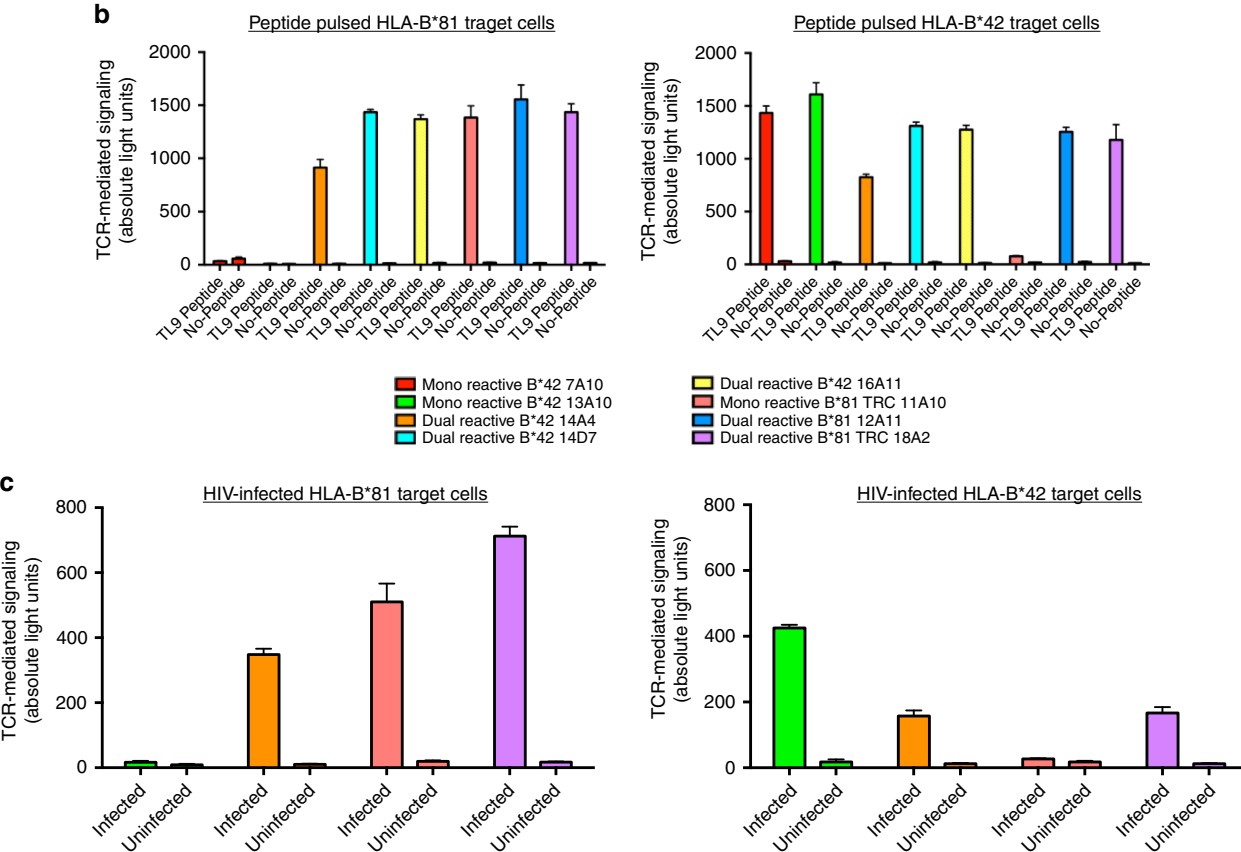

**Fig. 4** In vitro validation of TCR specificity and dual-reactivity. Details for the eight TL9-specific TCR clones investigated in this study are shown, including donor HLA, mono- or dual-reactivity phenotype, paired TCR α/β V gene usage and CDR3 sequences (**a**). Jurkat T cells were co-transfected with TCR α/β, CD8 α and an NFAT-driven luciferase reporter vector, and then co-cultured with TL9 peptide-pulsed (**b**) or HIV-infected (**c**) target cells stably expressing B*81:01 or B*42:01. TCR-dependent NFAT signaling was quantified by luminescence. The expected mono- or dual-reactive phenotype was observed for all reconstructed TCR clones, as indicated by greater luminescence (absolute light units, y-axis) in the presence of TL9-pulsed or virus-infected target cells compared to no-peptide or uninfected controls. Assays were conducted at least three times. Results from a representative experiment are shown as the mean of three co-culture reactions, plus standard deviation

function contributed to this phenotype. The three TCR clones isolated from B*81:01 expressing individuals (one mono- and two dual-reactive) displayed similar overall TL9 variant cross-recognition profiles, as demonstrated by Spearman R-values >0.80 for all pair-wise associations; however, greater breadth against variants at position 4 (aspartic acid) was seen for clone

18A2. In contrast, the five TCR clones isolated from B*42:01 expressing individuals displayed more disparate cross-recognition profiles, which was reflected by pair-wise Spearman R-values between 0.12 and 0.67. Notable differences were observed among B*42:01-derived TCR clones for recognition of TL9 peptide variants at positions 3 and 7, which are discussed below. To further

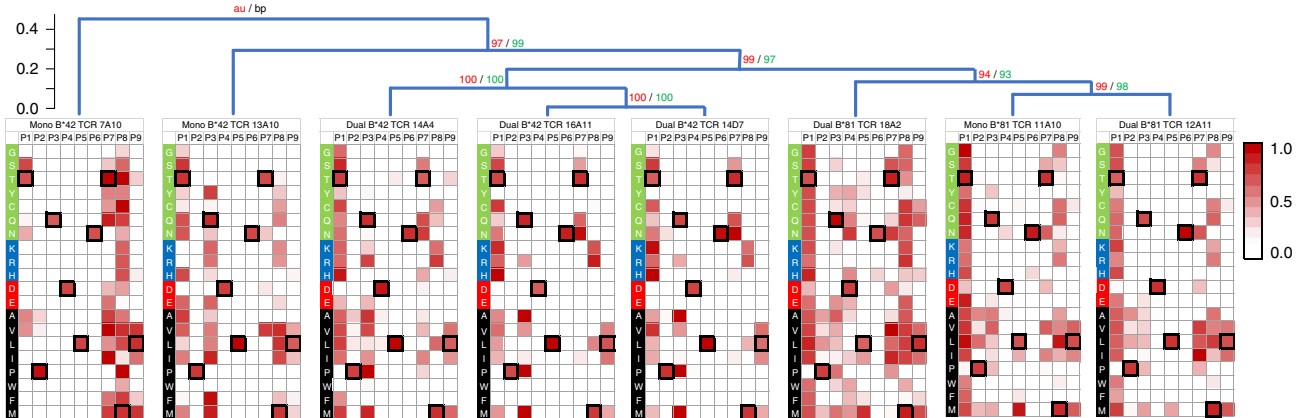

**Fig. 5** Functional clustering of TCR clones based on TL9 variant recognition profiles. TCR recognition of TL9 variants was assessed by pulsing target cells expressing the donor HLA (B*81:01 or B*42:01) with a panel of 180 peptides encompassing all single amino acid substitutions at epitope positions 1 through 9 prior to co-culture with Jurkat T cells expressing the TCR of interest. TCR-dependent NFAT signaling was quantified by luminescence. Values were normalized to the mean signal obtained for consensus TL9 (set to 1.0), which was tested nine times in each experiment. Results are displayed as heatmaps, where the warmer color reflects higher relative luminescence values indicative of better TCR recognition. Peptide positions are shown at the top of each heatmap; amino acid substitutions on the consensus TL9 backbone are shown on the left-hand side. Amino acids are grouped according to chemical properties: polar residues (G, S, T, Y, C, Q, N) are highlighted in green; basic residues (K, R, H) are blue; acidic residues (D, E) are red; and hydrophobic residues (A, V, L, I, P, W, F, M) are black. For reference, the consensus TL9 residue at each position is indicated using a box. TCR were grouped according to their functional profiles by hierarchical clustering using correlation distances and single linkage methods (5000 iterations) implemented in pvclust (http://stat.sys.i.kyoto-u.ac.jp/ prog/pvclust/). The dendrogram (top) displays approximately unbiased (au) p-values in red text and bootstrap probability (bp) values in green text. The three B*81:01-derived TCR clustered with bp values of 93 or higher. The three public dual-reactive B*42:01-derived TCR clustered with bp values of 100; and, notably, they grouped more closely with B*81:01 clones (bp value of 97), rather than mono-reactive B*42:01-derived TCR

evaluate the degree of functional similarity among these TCR clones, we performed a hierarchical clustering analysis based on their TL9 variant recognition profiles. Results are shown as a dendrogram in Fig. 5. We observed that all three of the public dual-reactive B*42:01-derived clones grouped together with bootstrap values of 100. Furthermore, this group of public clones clustered more closely with the three B*81:01-derived clones (bootstrap value of 97), compared to the two mono-reactive B*42:01-derived clones. Together, these results indicate that the epitope binding properties of the public dual-reactive B*42:01-derived TCR clones are more similar to those of clones elicited in the context of the more protective B*81:01 allele, despite TL9 peptide being presented on a different HLA allele.

**Dual-reactive TCR recognize more TL9 escape mutations**. We observed substantial differences in TL9 variant recognition among TCR clones, particularly at epitope positions 3 and 7. To examine the impact of these differences on viral adaptation, we restricted our analysis to 19 TL9 polymorphisms present in circulating HIV-1 subtype C sequences at a prevalence of ~0.1% or greater, which were considered as viable escape mutations. The ability of each TCR to recognize this panel of mutants is illustrated as a SequenceLogo in Fig. 6a–h. Collectively, the eight TCR clones recognized 16 (of 19, 84%) TL9 escape mutants; none responded to a threonine, glycine or aspartic acid substitution at position 3, which together accounted for 26.1% of circulating variant sequences. These results were highly consistent with prior studies of T cell cross-reactivity based on IFN-γ ELISPOT assays using PBMC[40,41,43], with 7 (of 8; 88%) TCR clones recognizing serine at position 7 or alanine at position 3, whereas responsiveness to other natural polymorphisms at position 3 (histidine, 13%; serine, 13%; threonine, 0%) and position 7 (valine, 75%; methionine, 25%) were less common.

While the total number of TL9 escape mutants recognized by B*81:01-derived TCR clones (range, 8–14), public dual-reactive B*42:01-derived clones (6–9), and mono-reactive B*42:01-derived clones (6–7) was not significantly different between groups, we observed that the two mono-reactive B*42:01-derived TCR clones responded primarily to variants located at either position 7 (for 7A10) (Fig. 6a) or position 3 (for 13A10) (Fig. 6b), indicating a limited ability to control escape mutations that occur at the other residue. In contrast, B*81:01-derived TCR clones and public dual-reactive B*42:01-derived clones each displayed broader recognition of variants at both position 3 and 7 (Fig. 6c–h), suggesting a distinct mechanism(s) of binding that accommodated changes at these residues. B*81:01-derived TCR clones also displayed broader recognition of TL9 variants at position 5. These results demonstrate functional differences among TCR clonotypes that may contribute to control of naturally occurring TL9 variants, particularly at epitope positions 3 and 7.

HIV-1 adaptation to CD8[+] T cells is highly dynamic[58,59], but escape in TL9 is limited by functional constraints[44]. Since common TL9 variants, such as serine at position 7 (S7; 23.4% of non-consensus sequences in LANL), are presumed to encounter a relatively lower barrier to escape compared to rare variants, such as isoleucine at this position (I7: 0.7%), we reasoned that TCR recognition of more common TL9 variants would be beneficial for viral control. To explore this, we plotted these 19 TL9 polymorphisms using pie charts with wedges sized according to their prevalence in subtype C sequences (Fig. 6i–p) and then determined the percent coverage of TL9 escape mutations for each TCR by calculating the proportion of total sequence variation that was recognized (see wedges highlighted in Red). Based on this frequency-adjusted analysis, individual clones displayed 22% (Fig. 6j) to 67% (Fig. 6p) coverage of TL9 variants. While differences between groups were not statistically significant, B*81:01-derived TCR clones and public dual-reactive

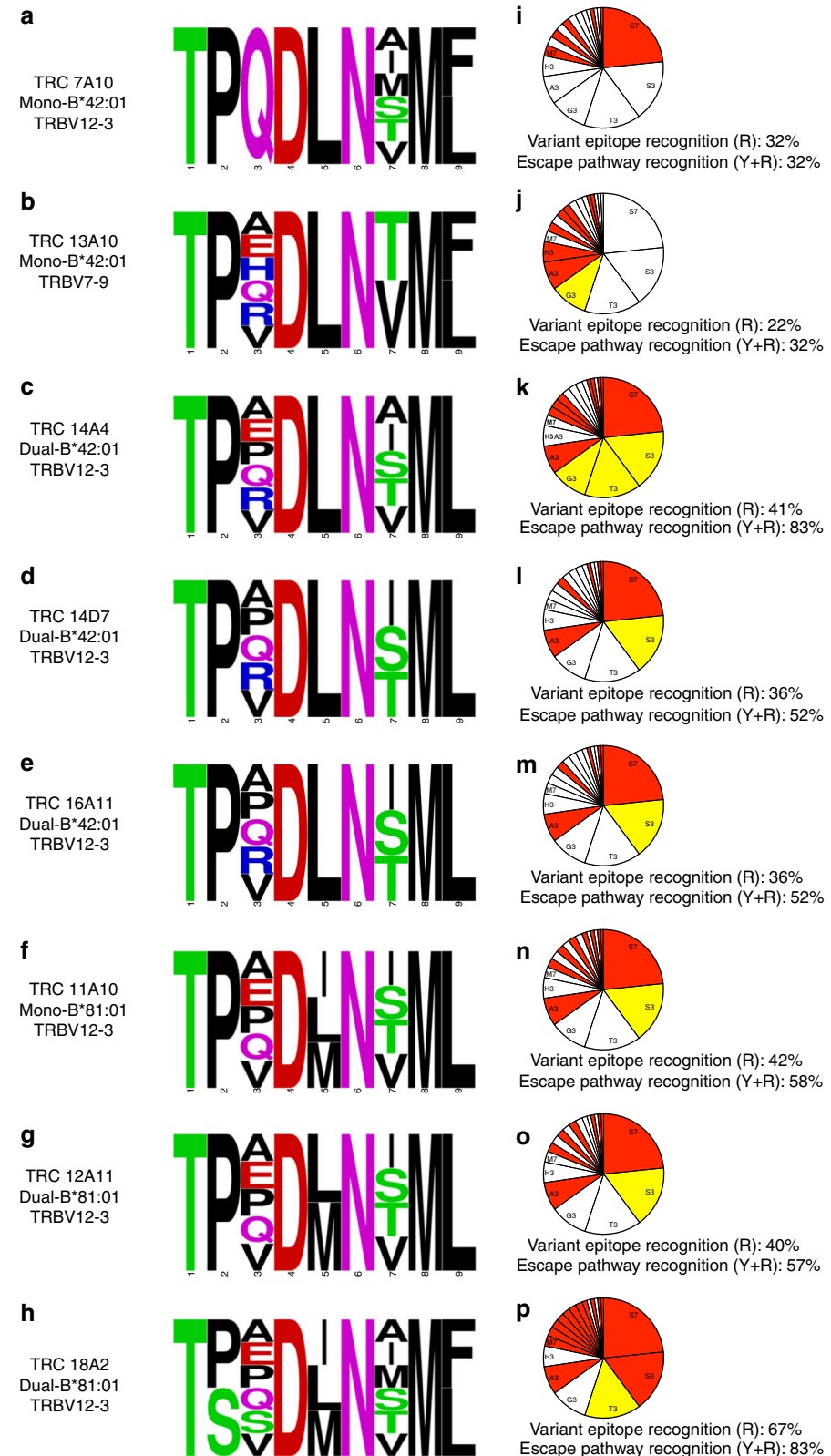

B*42:01 clones tended to display greater coverage (40–67% and 36–41%, respectively) compared to the mono-reactive B*42:01 clones (22 and 32%). These results indicate that individual TCR clonotypes display variable capacity to recognize more common TL9 variants that are likely to constitute preferential escape mutants.

B*42:01 clones tended to display greater coverage (40–67% and 36–41%, respectively) compared to the mono-reactive B*42:01 clones (22 and 32%). These results indicate that individual TCR clonotypes display variable capacity to recognize more common TL9 variants that are likely to constitute preferential escape mutants.

Since codon usage places additional constraints on viral sequence evolution, we reasoned that TCR recognition of transitional variants would hinder the development of TL9 escape. For example, substitution of glutamine at position 3 (Q3) with serine (S3) requires a minimum of two nucleotide changes with transition through proline (P3) or a stop codon. Thus, TCR

**Fig. 6** Enhanced recognition of TL9 escape by B*81:01-derived and B*42:01-derived dual-reactive TCR clones. The ability of each TCR clone to respond to HIV-1 escape mutants was determined by comparing its recognition profile to a panel of 19 naturally occurring subtype C TL9 variants, found at a prevalence of ~0.1% or greater in the LANL HIV Sequence Database (HIV Databases; http://www.hiv.lanl.gov). Recognition breadth for each TCR is illustrated as a SequenceLogo, demonstrating variable responsiveness toward relevant TL9 mutations located primarily at epitope positions 3 and 7. Mono-reactive TCR from B*42:01 expressing individuals displayed narrower profiles that recognized TL9 variants at either position 7 (7A10, **a**) or position 3 (13A10, **b**), whereas public dual-reactive B*42:01-derived clones (**c–e**) and B*81:01-derived clones (**f–h**) demonstrated broader ability to recognize TL9 variants at both positions 3 and 7. To account for constraints on TL9 escape, epitope variants were displayed using pie charts where the size of each wedge is proportional to variant frequency in circulating subtype C isolates (**i–p**). Serine at position 7 (S7), serine at position 3 (S3), and threonine at position 3 (T3) accounted for the majority (55%) of population-level variation. For each chart, the wedge is shaded in red (R) if the TCR responded to the escape mutant or in yellow (Y) if the TCR recognized all transitional mutations required to generate that escape mutant from consensus TL9. The sum of all red wedges is displayed under each chart as the total percentage of Variant Epitope Recognition and the sum of all shaded wedges, red plus yellow, is displayed as the total percentage of Escape Pathway Recognition, where recognition of all circulating TL9 variants would be 100%. Overall, B*81:01-derived TCR (**n–p**) and public dual-reactive B*42:01-derived TCR (**k–m**) displayed better ability to cross-recognize circulating TL9 escape variants and pathways compared to mono-reactive B*42:01-derived clones (**i–j**)

recognition of the P3 variant would be expected to prevent formation of S3, even in cases where the TCR did not respond to the S3 variant itself. For each TCR clone, we determined which TL9 escape mutations were inhibited due to recognition of critical transitional variants (see yellow wedges in Fig. 6i–p). We then calculated the total coverage of TL9 escape pathways by summing the proportion of variant sequences that were recognized or prevented by each TCR (i.e., red plus yellow wedges). Based on this pathway-adjusted analysis, individual TCR clones displayed 32–83% coverage of TL9 escape mechanisms. Notably, B*81:01-derived clones (range, 57–83%, Fig. 6n–p) and public dual-reactive B*42:01-derived clones (range, 52–83%, Fig. 6k–m) displayed broader coverage compared to mono-reactive B*42:01-derived clones (both 32%, Figs. 6i–j) ($p = 0.05$ and $p = 0.11$, respectively; Student's T test); highlighted by one public dual-reactive B*42:01-derived clone (14A4) and one B*81:01-derived clone (18A2). Notably, extended coverage of TL9 escape pathways was due mainly to the ability of TCR clones 14A4 (**K**), 14D7 (**L**), 16A11 (**M**), 11A10 (**N**), and 12A11 (**O**) to respond to the P3 variant, which is anticipated to impair development of the S3 escape mutation that accounts for 16.5% of TL9 variant sequences. Together, these results illustrate the functional diversity that exists among antigen-specific T cells and demonstrate the impact of TCR sequence on recognition of HIV-1 Gag TL9 escape mutations. This work highlights the role of TCR clonotype differences as a correlate of HIV-1 control in the context of HLA B*81:01 and B*42:01.

## Discussion

The characteristics that determine effectiveness of adaptive host immune responses to rapidly evolving pathogens such as HIV-1 are not fully defined. In this study, we examined the CD8+ T cell response against the immunodominant HIV-1 p24 Gag TL9 epitope in the context of two closely related class I HLA alleles, B*81:01 and B*42:01, that display differential abilities to control viral subtype C infection[9]. We identified a population of dual HLA tetramer-reactive T cells that recognized TL9 presented in the context of either B*42:01 or B*81:01 alleles and observed that the presence of this dual-reactive population was an independent predictor of lower plasma viral load. In B*42:01 expressing individuals, dual-reactive populations were dominated by public TCR clonotypes that encoded *TRBV12-3*. A comprehensive in vitro functional analysis of selected TCR clones indicated that B*81:01-derived clones (regardless of mono- or dual-reactive phenotype) and public dual-reactive B*42:01-derived TCR clones displayed greater ability to recognize TL9 escape pathways, compared to mono-

reactive clones from B*42:01 expressing individuals. While the dual-reactive T cell phenotype reported here is a phenomenon of tetramer binding to pHLA that is not expressed by the host, our results indicate that it identifies T cell subsets within diverse antigen-specific repertoires that share important features, including Vβ gene sequences and the ability to recognize HIV-1 epitope variants. A similar dual HLA-reactive phenotype has been described for one CTL clone[51], but here we demonstrate the extent to which dual-reactive T cells exist in vivo and link this phenotype to functional characteristics of individual TCR clonotypes. It will be critical to examine this phenomenon further to see if it is a common feature of T cell responses elicited in the context of other HLA supertypes, such as members of the B57 family that also show differential abilities to control HIV-1 infection[9].

It is important to note that antigen sensitivity appeared to be independent of cross-reactivity for TCR examined in this study. While more detailed biochemical analyses will be necessary to fully assess the affinity of TL9-specific TCR clones, our reporter assay provides a surrogate measure of antigen sensitivity based on strength of NFAT signaling. The activities of dual-reactive B*42:01-derived TCR clones were lower compared to those of mono-reactive B*42:01-derived clones. We observed similar differences in the sensitivity of representative TCR clones tested over a range of TL9 concentrations (Supplementary Fig. 3), indicating that this result was not an artifact of peptide dose. In addition, TCR sensitivity toward consensus TL9 did not correlate with cross-recognition of TL9 variants in our more comprehensive analysis, although it will be important to confirm this observation using a larger panel of TL9-specific TCR clones.

Although B*81:01 and B*42:01 are both members of the B7 supertype and known to present many of the same HIV-1 peptides, the dual-reactive T cell phenotype is unexpected since structural data indicated that TL9 adopts a distinct conformation upon binding to each allele[45]. Our analysis demonstrated that B*81:01-derived TCR clones and public dual-reactive B*42:01-derived clones recognized TL9 variants at both principal sites of viral escape, positions 3 and 7. This is interesting since both residues are buried in the B*81:01 structure, while position 7 is solvent-exposed in the context of B*42:01[45]. It remains to be determined whether TCR recognition reflects direct binding to these TL9 variants, or rather is due to conformational changes in the pHLA or indirect effects on other TL9 residues. In contrast, mono-reactive B*42:01-derived TCR displayed breadth against TL9 variants at either position 3 or position 7, but not both. While both types of mono-reactive TCR may be present within the repertoire of B*42:01 expressing individuals, skewing of the

immune response toward either mono-reactive TCR subset could facilitate viral escape at the alternative TL9 position.

Our detailed functional data provides insight into characteristics of TL9-specific TCR clones that might be overlooked using more conventional methods based on HIV-1 sequences alone. For example, all TCR clones were sensitive to changes at TL9 position 6, demonstrating that this highly conserved polar asparagine residue is critical in the context of both HLA alleles, despite it being solvent-exposed in the B*81:01 structure and buried in the B*42:01 structure[45]. In addition, most TCR were sensitive to changes at position 4, indicating that this negatively charged, polar aspartic acid residue (which is solvent-exposed in both structures[45]) is critical for recognition; however, the B*81:01-derived clone 18A2 tolerated mutations at this residue, suggesting a distinct mechanism of interaction in this case. Structural flexibility is a crucial feature of the interaction between TCR and pHLA;[60–62] thus, changes in conformation induced upon TCR binding may be relevant to recognize TL9 variants in the context of both B*81:01 and B*42:01. Because such conformational rearrangements are difficult to predict[63], more detailed structural analyses will be necessary to explore this issue. In the absence of such data, we are unable to define structural determinants of cross-reactivity for the TCR clones examined in our study. Nevertheless, our results highlight peptide-recognition properties that may contribute to future studies of these and other TL9-specific TCR.

This work extends prior efforts to examine TL9-specific CD8+ T cell responses. In particular, Leslie et al.[40] and Geldmacher et al.[41] observed enrichment of *TRBV12-3* usage in B*81:01 and some B*42:01 expressing individuals, while Leslie et al.[40] and Kloverpris et al.[34] described public TCR β sequences in B*42:01 expressing individuals that correspond to the dual-reactive TCR clones 14A4 (CASSFSKNTEAFF) and 14D7 (CASSHSKNTEAFF) examined here and demonstrated that the presence of these public clones was associated with TL9 immunodominance[34]. These earlier reports suggested that CD8+ T cell responses in B*81:01 expressing individuals displayed broader recognition of TL9 variants, but individual T cell clones (or TCR clonotypes) were not explored. Here, we re-discovered public TCR β clonotypes in B*42:01 expressing individuals by their dual HLA-reactive phenotypes. Extensive functional analyses of selected TCR clones demonstrated substantial diversity in their abilities to recognize TL9 variants. Our results emphasize the role of cross-reactive public TCR clones encoding *TRBV12-3* for effective TL9 responses in B*42:01 expressing individuals; however, differences in TL9 variant recognition among these public clones also suggests a functional hierarchy that may be clinically relevant. Furthermore, it should be noted that B*42:01-derived clone 7A10 encoded *TRBV12-3* but did not demonstrate dual-reactivity or broad recognition of TL9 escape variants, indicating that phenotypic differences among TCR were not driven entirely by V gene usage.

Several observations from this study are relevant for the design of vaccines or therapeutics. Vaccine antigens that can elicit effective cross-reactive TCR clonotypes might provide better protection against HIV-1 infection or enhance the ability of the immune system to recognize latent viral reservoirs encoding escape variants. We observed that public dual-reactive TCR clones from B*42:01 expressing individuals were unique in their ability to recognize a proline variant at TL9 position 3 (Q3P). It would be interesting to examine ex vivo responses to this rare TL9 variant as a surrogate marker for public dual-reactive T cells HIV-infected individuals or vaccine recipients; or to consider vaccination with this variant TL9 sequence to elicit a more broadly reactive T cell response in B*42:01 expressing individuals. We have also identified and validated the recognition profiles of

eight TL9-specific TCR clones, including several with dual HLA-reactive phenotypes. These TCR clones may be attractive products for future T cell therapy strategies that aim to reduce or eliminate viral reservoirs encoding escape mutations in the context of HIV-1 subtype C infection.

In summary, we have identified characteristics of TCR clonotype sequence and function that are associated with variable control of HIV-1 infection in the context of B*81:01 and B*42:01. We observed a unique dual HLA-reactive CD8+ T cell population that was highly enriched for a small number of public TCR clonotypes in B*42:01 expressing individuals. Mono- and dual-reactive TCR clones from individuals expressing the protective B*81:01 allele displayed broad recognition of TL9 variants, suggesting that they provide comparable abilities to contain HIV-1 Gag escape mutants. In contrast, only public dual-reactive TCR clones from B*42:01 expressing individuals displayed similar broad TL9 variant recognition, suggesting that these public clonotypes provide enhanced ability to control HIV-1 escape mutants in the context of this less protective HLA allele. While additional studies will be necessary to fully assess the structural mechanisms and clinical relevance of these observations, this work provides a strong foundation and rationale to further explore the impact of TCR clonotype differences on HIV-1 outcomes. Together, our results highlight the feasibility and use of detailed molecular analyses that link TCR sequences with functional characteristics to improve understanding of T cell responses against diverse and rapidly evolving pathogens. Similar investigations might be beneficial to enhance the development of vaccines and T cell-based immunotherapies against HIV or other human diseases.

## Methods

**Study subjects**. Twenty-one antiretroviral naive individuals were enrolled in Durban, South Africa through the HIV Pathogenesis Program (HPP) acute infection cohorts. The clinical characteristics are shown in Table 1. All individuals were infected with HIV-1 subtype C. The Biomedical Research Ethics Committee of the University of KwaZulu-Natal and the Massachusetts General Hospital Ethics committee approved this study. All subjects provided written informed consent.

**HLA typing**. HLA typing was conducted by the laboratory of Dr. Mary Carrington (National Cancer Institute, Fredrick, USA), as previously described[9]. DNA samples obtained from peripheral blood mononuclear cells (PBMC) were first oligo-typed using Dynal RELITM reverse Sequence Specific Oligonucleotide (SSO) kits for the HLA-A, HLA-B, and HLA-C loci (Dynal Biotech). Genotypes were refined to the allelic level using the Dynal Biotech Sequence Specific priming (SSP) kits in conjunction with the previous SSO type. In cases where alleles were still not well-defined at the allelic level, sequence-specific primers were used[64]. All class I HLA alleles in the IMGT allele release 24.0 were considered in the typing.

**Tetramer staining, cell sorting, and cell line generation**. To identify and characterize TL9-specific CD8+ T cell populations, PBMC were first stained with a cell-viability dye (Fixable Blue Dead Cell Stain Kit, Invitrogen) for 10 min at room temperature. Cells were washed with 2% fetal calf serum (FCS) in phosphate buffered saline (PBS) and then stained with B*42:01-APC and/or B*81:01-PE TL9 HLA class I tetramers (obtained from the laboratory of Dr. Soren Buus), for 30 min at room temperature. Subsequently, cells were washed, and surface stained with anti-CD8-BV786, CD3-BV711, and CD4-BV650 for 20 min at room temperature. Stained cells were analyzed by flow cytometry and/or tetramer-specific CD8+ T-cells were sorted for TCR sequencing. To generate TL9-specific CD8+ T-cell lines, cells were pulsed with 5 μl (200 μg ml$^{-1}$) of TL9 peptide at 37 °C for 3 h and subsequently cultured in RPMI medium containing 10% heat-inactivated fetal calf serum (R10 medium) supplemented with 50 units ml$^{-1}$ of recombinant human interleukin 2 (IL-2) (R10/50 medium) for 2 weeks. Expanded TL9-specific CD8+ T cells were validated for specificity by tetramer staining and isolated using a cell sorter (BD FACSAria, Germany).

**Tetramer intracellular cytokine staining and ELISPOT assay**. To assess the functional quality of TL9-specific CD8+ T cells, PBMC from B*81:01 and B*42:01 subjects were stimulated with 1.2 μl (200 μg ml$^{-1}$) of TL9 peptide for 6 h.

After stimulation, cells were stained with an equal mixture of B*81:01 and B*42:01 TL9 tetramers for 30 min at room temperature, washed in PBS containing 2% FCS and then stained with viability dye, anti-CD8-BV786, CD3-BV711, and CD4-BV650 for 20 min at room temperature. Cells were fixed, permeabilized, stained intracellularly with anti-IFN-γ-PE-Cy7 and analyzed on the BD LSRFortessa. HIV-1 immune responses were enumerated by IFN-γ enzyme-linked immunosorbent spot (ELISPOT) assay as previously described[9,65]. Briefly, PBMCs were stimulated with optimal HIV-1 subtype C peptide corresponding to each patient's HLA-A, B, and C alleles at a final concentration of 2 μg ml$^{-1}$ peptide.

**TCR Vβ antibody staining**. TCR variable β (Vβ) expression on mono- and dual-tetramer$^+$ cells was assessed by flow cytometry as described previously[66]. PBMC were stained with B*81:01 and/or B*42:01 TL9 tetramers conjugated to different fluorochromes, followed by TCR Vβ family labeling using IOTest Beta Mark TCR Vβ repertoire kit (Beckman Coulter, Pasadena, USA) for 30 min at room temperature. Subsequently, cells were stained with viability dye, anti-CD8-BV786, CD3-BV711, and CD4-BV650. The percentage of each Vβ family was determined for a minimum of 100,000 CD8$^+$ T cells using FlowJo software (Treestar, Ashland, USA). TCR Vβ staining was also performed on expanded mono- and dual-TL9 tetramer$^+$ cell lines.

**TCR sequencing**. Amplification of TCR β CDR3 coding regions from single T cells was performed as described previously by Han et al.[67] with modifications to obtain ~230 bp amplicons for Sanger sequencing (ABI 3130xl). Primers are included in Supplementary Tables 2 and 3. The one-step SuperScript III kit (ThermoFisher) was used for RT-PCR and Expand High Fidelity PCR system (Roche) was used for subsequent rounds. TCR α amplicons were TOPO cloned and screened to ensure productive CDR3 rearrangement. Sequences were examined using the ImMunoGeneTics (IMGT)/V-quest tool (www.imgt.org) to characterize Variable gene usage and CDR3 diversity. Full-length TCR alleles were reconstructed using Variable and Constant gene sequences obtained from the IMGT database, codon-optimized using the CodonOpt tool (Integrated DNA Technologies; www.idtdna.com) and synthesized as double-stranded DNA gBlocks by IDT. Full-length genes were cloned into pSELECT_GFPzeo (Invivogen) for functional studies.

**TCR reporter assay**. TCR antigen recognition was examined using a previously described in vitro reporter T cell assay[56]. Briefly, Jurkat T cells were co-transfected with TCR α, TCR β, CD8 α, and NFAT-driven luciferase reporter plasmids by electroporation (BioRad MxCell). Target cells consisted of a CEM-derived GXR cell line[68] stably expressing either HLA-B*42:01 or B*81:01. TCR-transfected Jurkat effector cells (50,000 cells) were co-cultured with 50,000 target cells either pulsed with 20 μM TL9 peptide (purchased from GenScript at >90% purity) or infected with HIV-1 in a total volume of 100 μl, and TCR recognition activity was quantified by luminescence after 6 h (Tecan M200). Viral stocks were generated by co-transfection of HEK293T cells with pBR4.3ΔNefΔEnv and pVSV-g using Lipofectamine 2000 (ThermoFisher Scientific). Infected GXR target cells were isolated by FACS based on GFP expression prior to co-culture with Jurkat T cells. To screen antigen cross-recognition, a peptide panel consisting of all single amino acid TL9 variants (180 total peptides) was purchased from GenScript. This panel was prepared using microscale synthesis methods and individual peptides were aliquoted to 96-well plates at 0.5–2.0 mg total weight and >75% purity. Target cells were pulsed with ~20 μM peptide; however, due to variations in peptide sequence, total weight, molar weights and purity, actual concentrations were anticipated to range between 8.4 and 20 μM.

**HIV sequence analysis**. To determine the frequency of naturally occurring Gag TL9 variants in HIV-1 subtype C infection, all subtype C TL9 amino acid sequences (N = 5481) were downloaded from the Los Alamos National Laboratory (LANL) HIV sequence database (www.hiv.lanl.gov) and analyzed. Sequences encoding consensus TL9 (N = 4526), multiple substitutions or mixed residues (combined N = 217) and those appearing fewer than five times (N = 53) were removed to generate a list of the most probable single amino acid TL9 variants (N = 685). The proportion of each variant within this population was calculated to determine the likelihood of viral escape. Critical transition mutations (i.e., those that must occur for consensus TL9 to evolve into escape variants) were identified using the standard amino acid codon table for eukaryotes. Sequence conservation frequencies for TL9 residues were estimated using the QuickAlign tool on the LANL web site (based on N = 1865 protein sequences).

**Statistical analysis**. Statistical analyses were conducted using Prism software, version 6.0 (GraphPad, Inc.). Two-tailed tests were employed, and p-values <0.05 were considered to be significant. Comparisons between groups of continuous variables were assessed using parametric (unpaired Student's T) or non-parametric (Mann–Whitney U) tests. Differences in categorical variables between groups were assessed using Fisher's exact test. A multivariable linear regression analyses was conducted using Stata, version 14 (Stata Corp), to assess the independent predictive ability of HLA and dual-reactivity on plasma viral loads. Hierarchical clustering analysis was performed using pvclust software[69] (http://stat.sys.i.kyoto-u.ac.jp/prog/pvclust), implemented in R. Data was grouped according to correlation

distances using single linkage methods. Approximately unbiased (au) p-values and bootstrap probability (bp) values were based on 5000 iterations.

## Data availability
Participants specimens, primary cell data, reagents for TCR sequencing and reporter cell assays used for peptide screening are available from the corresponding authors on reasonable request. Contact Zaza M. Ndhlovu (zndhlovu@mgh.harvard.edu) with requests regarding reagents and resource sharing for participant specimens, sequences and primary cell data. Similar requests regarding TCR sequencing and reporter cell assays used for peptide screening should be directed toward Mark A. Brockman (mark_brockman@sfu.ca). Paired TCR sequences are available through Genbank, accession codes: [MH918759-MH918774].

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

## Acknowledgements

This work was supported in part through the Sub-Saharan African Network for TB/HIV Research Excellence (SANTHE), a DELTAS-Africa Initiative (Grant # DEL-15–006). The DELTAS-Africa Initiative is an independent funding scheme of the African Academy of Sciences's Alliance for Accelerating Excellence in Science in Africa and supported by the New Partnership for Africa's Development Planning and Coordinating Agency with funding from the Wellcome Trust [Grant # 107752/Z/15/Z] and the Government of the United Kingdom. Additional funding was received from the National Institutes of Health, USA (R37-AI080289, R01-AI102660 and UM1-AI126617), the International AIDS Vaccine Initiative (UKZNRSA1001), and the Canadian Institutes for Health Research (HIG-133050). Z.M.N. is supported by an HHMI International research scholar award (Grant #55008743); M.A.B. is supported by the Canada Research Chairs Program and T.N. is supported by the South Africa Research Chairs Initiative. We thank Dr. Bruce Walker for providing clinical samples, part of the funding and for his insightful discussion during preparation of this manuscript. We also thank Natalie Kinloch, Xiaomei Kuang, and Zabrina Brumme for technical assistance and helpful discussions during preparation of this manuscript. The views expressed in this publication are those of the authors and not necessarily those of the funding agencies or partner organizations.

## Author contributions

F.O. and G.A. conducted the experiments and analyzed data under the supervision of Z.M.N. and M.A.B. R.M., E.G., T.Nk., D.M., J.M., N.I. and D.C. provided technical support for experiments or data analysis. Z.M.N., T.N. and M.A.B contributed to the

experimental design and data analysis. F.O., G.A., M.A.B., and Z.M.N. wrote the manuscript and all authors contributed to revisions.

## Additional information

**Competing interests:** The authors declare no competing interests.

