## [Peer Review File · Nature Communications]

Reviewers' comments:

Reviewer #1 (Remarks to the Author):

The authors have performed a comprehensive evaluation of TCR usage in individuals with the HLA B*42 and B*81 alleles. They identified dual HLA restriction as an independent correlate of control of viremia. This dual HLA restriction was associated with public TCR clonotype usage and broader recognition of variant epitopic peptides. I have several questions/suggestions that may increase the overall impact of the manuscript.

Table: Only HLA data for B*42 and B*81 are shown (and individuals with B*5703, B*5801, B*3901 were excluded). Either in this table, or in a separate table, the authors should show the full class I HLA type.

Are any of the individuals B*0702? This allele also binds the TL9 epitope.

The authors state "The p24 Gag-derived epitope TL9 is immunodominant in both B*81:01 B*42:01 expressing individuals". Immunodominant can mean that the majority of individuals with these alleles recognize this epitope. It can also mean that when this epitope is recognized, it is the highest magnitude response. The authors should show how this response compares to other HLA class I responses in this cohort.

In individuals with dual-reactive populations, was the magnitude of the response higher than responses restricted by other alleles? It would be interesting to mention whether dual reactive populations are associated with immunodominance of the response.

Reconstructed TCR were used in a Jurkat system to evaluate recognition of 180 amino acid variants. What peptide concentrations were used to evaluate recognition? Were peptide dilutions evaluated for recognition? This peptide and variants have been extensively studied; it would be useful to know how recognition of variants in the Jurkat system compare to previously evaluated T cell clones, or even PBMC. More specifically, how does the antigen sensitivity of dual reactive TCR to the parent and variant peptides compare to that of single reactive TCR? It may be the case that higher antigen sensitivity to the parent epitope predicts cross-reactivity.

Reviewer #2 (Remarks to the Author):

This manuscript describes an interesting and potentially important cross recognition of CD8 T cells specific for the HIV-1 Gag peptide TL9 restricted by HLA-B*81 and HLA-B*42. B*81 is a protective HLA type, associated with lower virus load and reduced escape; B*42 presents the same peptide but is not protective. The structures of both HLA molecules with this peptide have been determined and there are striking differences in the orientations of side chains 5, 6 and 7 between the two HLA molecules.

They show here that B*81-TL9 specific TCRs are restricted, using predominantly BV12-3 though with variable CDR3 regions (I assume this is what the various grey pie segments means in figure 3). In both B*42 and B*81 donors, there are TCRs that recognize both B*81 and B842 bound to the peptide, and these also use BV12-3. In the latter case, there are large single clones – are these identical to the large single clones in the B81 responses in figure 3?

The cross reactivity is puzzling in view of the different structures in the Kloverpris et al paper in Retrovirology. They show here the importance to T cell recognition of peptide residue 6 which is orientated very differently in the two structures. It seems most likely therefore that the peptide

flips into the B*81 orientation when the TCR binds, but that would need structural confirmation

They go on to test cloned TCRs of the three types on cells pulsed with 180 variant peptides, covering all changes at each of the nine amino acids. It would be helpful if they could test each of these for binding to the two HLA molecules so that the non-binders can be discriminated from those that only interact with the TCR. Also they should indicate what peptide concentration was used; it would be better if these experiments were done on a minimum of three peptide concentration, as high concentrations are likely to minimize the differences and low concentrations would enhance them. That might be too difficult to do for all 180 variants but at least the peptide should be titrated for the wild-type sequence and they should show that they chose a suitable concentration to test the variant peptides. Similarly the peptide concentrations used in the experiment shown in figure 6 should be given and titrated.

A complete story would show the structures of the critical TCRs complexed to B*81 and B*42. That is a lot to ask but it is the obvious question and it is possible that they are already doing this. If so, waiting to include these data would make the paper much stronger. Without this information it we are left a bit in the air as to how all this works. If the TCR flips the peptide conformation, I think that would be a 'first', although Tynan, Rossjohn et al have described a TCR flattening a bulging peptide (Nature Immunology 2007).

The overall conclusion is that more broadly cross reactive T cells are better and control virus better, limiting escape. It is important that when patients are divided into those that do or do not they express cross reactive receptors, virus loads were significantly lower in the former, possibly better than the B*81 protection. The fact that both B81 and B42 can elicit such receptors, but that B42 does so less efficiently (presumably because of the different peptide orientation) is intriguing.

Reviewer #1 (Remarks for the Author):

The authors have performed a comprehensive evaluation of TCR usage in individuals with the HLA B*42 and B*81 alleles. They identified dual HLA restriction as an independent correlate of control of viremia. This dual HLA restriction was associated with public TCR clonotype usage and broader recognition of variant epitopic peptides. I have several questions/suggestions that may increase the overall impact of the manuscript.

Table: Only HLA data for B*42 and B*81 are shown (and individuals with B*5703, B*5801, B*3901 were excluded). Either in this table, or in a separate table, the authors should show the full class I HLA type.

We thank the reviewer for this suggestion. We have added a new data (Table 2) that reports full class I HLA allele profiles of our study participants and we have revised the text in line 114 and 115 to reflect these changes.

Are any of the individuals B*0702? This allele also binds the TL9 epitope.

As now indicated in Table 2, none of our study participants expressed B*07:02.

The authors state “The p24 Gag-derived epitope TL9 is immunodominant in both B*81:01 B*42:01 expressing individuals”. Immunodominant can mean that the majority of individuals with these alleles recognize this epitope. It can also mean that when this epitope is recognized, it is the highest magnitude response. The authors should show how this response compares to other HLA class I responses in this cohort.

We thank the reviewer for this suggestion. We have now included a new supplementary figure 1 showing that TL9 responses were greater than other responses within individuals. We have added a sentence in the in line 130-131 to clarify this issue.

In individuals with dual-reactive populations, was the magnitude of the response higher than responses restricted by other alleles? It would be interesting to mention whether dual reactive populations are associated with immunodominance of the response.

The reviewer makes a valid point on the magnitude of dual-reactive populations compared to other alleles. The dual-reactive response was not higher than the responses restricted by other alleles except in 2 B*81:01 participants. We think that the significance of dual-reactivity is in the quality rather than the quantity of response.

Reconstructed TCR were used in a Jurkat system to evaluate recognition of 180 amino acid variants. What peptide concentrations were used to evaluate recognition?

We apologize for omitting this crucial information from our manuscript. Initial TCR reporter assays (shown in Figure 4) were conducted using 20 μ M TL9 peptide (see Methods, line 521). This concentration is comparable to prior studies of CTL recognition based on IFN- γ Elispot assays (e.g. Leslie et al, J Immunol 2006), which typically screen cells using peptide concentrations ranging from 10 to 200 μ M.

The panel of 180 TL9 peptide variants was prepared using a “microscale” synthesis product (purchased from GenScript) that provided each peptide at 0.5 to 2.0 mg total weight and >75% purity. These assays (shown in Figures 5 and 6) were also conducted using ~20 μ M peptide; however, due to variations in peptide sequence, total weight, Molar weight, and purity, we estimate that individual peptides were tested within an actual range of 8.4 to 20 μ M. We note that recognition of consensus TL9 was highly reproducible, despite being synthesized independently nine times within the variant peptide panel. Also,

while dose-dependent differences in TCR-mediated signalling were observed for TL9 (see Figure below), these differences were relatively modest (< 2-fold) at concentrations above 1 μ M. These details are now discussed in the Methods (lines 518-523) of the revised manuscript.

Were peptide dilutions evaluated for recognition?

We examined TCR recognition using serial dilutions of consensus TL9 peptide (see Figure below) and now report these results as “data not shown” on lines 209-212 of the revised manuscript. We observed similar mono- or dual-reactivity at peptide concentrations ranging from 5 nM to 20 μ M, demonstrating that TCR phenotypes are not an artefact of peptide dose. Notably, TCR-mediated signalling was observed at lower peptide doses when B*81-expressing target cells were used, which is consistent with a prior report from Leslie et al (J Immunol 2006) showing higher antigen sensitivity in the context of this HLA allele.

Supporting Figure: TCR signalling in response to TL9 peptide dilutions. Mono-reactive TCR clones 11A10 (B*81; red) and 13A10 (B*42; green) and dual-reactive TCR clone 14A4 (B*42; orange) were tested using target cells expressing B*42 (left panel) or B*81 (right panel). Similar mono- or dual-reactive phenotypes were observed over a wide range of TL9 peptide doses (5 nM to 20 μ M). Both mono-reactive clones displayed greater signalling activity compared to the dual-reactive clone at all peptide doses. Combined with data shown in Figure 4, results suggest that antigen sensitivity is independent of dual-reactivity.

This peptide and variants have been extensively studied; it would be useful to know how recognition of variants in the Jurkat system compare to previously evaluated T cell clones, or even PBMC.

While TL9 has been extensively studied, prior reports typically employed IFN- γ Elispot assays using PBMC and focused on a small number of natural HIV epitope variants, including those that differ between viral subtypes. Leslie et al (J Immunol 2006) or Geldmacher et al (AIDS 2007; Blood 2009) examined TL9 variants S7, M7, V7, T3, S3, A3, and H3. While both groups observed substantial variation among participants, B*81-expressing individuals tended to display greater cross-recognition compared to B*42-expressing individuals. TL9 variants S7 and A3 tended to be recognized most frequently in both B*81:01 and B*42:01 participants, particularly in cohorts infected with subtype C strains. Our results are consistent with these reports. In particular, variants S7 and A3 were each recognized by 7 of 8 TCR clones tested, while cross-recognition was less common

for M7 (2 of 8 clones), V7 (6/8), T3 (0/8), S3 (1/8), and H3 (1/8). This point is now discussed in our revised text (see **lines 260-264**).

We know of no prior reports that assessed TL9 variant recognition using CTL clones, so this comparison is not possible. Furthermore, since we did not directly compare the specificity of T cell clones to isolated TCR in our study, we cannot address this issue with data. This question will be a priority for future research in this area.

More specifically, how does the antigen sensitivity of dual reactive TCR to the parent and variant peptides compare to that of single reactive TCR? It may be the case that higher antigen sensitivity to the parent epitope predicts cross-reactivity.

While more detailed biochemical studies are necessary to assess the affinity of TL9-specific TCR clones, our luciferase reporter assay provides a surrogate measure of antigen sensitivity based on strength of NFAT signalling. As shown in Figure 4 and the supporting figure provided in response to Reviewer 1 (above), the activities of B*42-derived dual-reactive TCR (14A4, 14D7, 16A11) were lower compared to those of mono-reactive clones (7A10, 13A10). These results suggest that cross-recognition of TL9 presented on B*81 is independent of antigen sensitivity for these TCR. In addition, TCR sensitivity towards consensus TL9 did not correlate with cross-recognition of TL9 variants in our comprehensive analysis; however, it will be important to confirm this result using a larger panel of TL9-specific TCR clones.

We have revised the text of our manuscript to discuss these points (see **lines 342-352**).

Reviewer #2 (Remarks for the Author):

This manuscript describes an interesting and potentially important cross recognition of CD8 T cells specific for the HIV-1 Gag peptide TL9 restricted by HLA-B*81 and HLA-B*42. B*81 is a protective HLA type, associated with lower virus load and reduced escape; B*42 presents the same peptide but is not protective. The structures of both HLA molecules with this peptide have been determined and there are striking differences in the orientations of side chains 5, 6 and 7 between the two HLA molecules.

They show here that B*81-TL9 specific TCRs are restricted, using predominantly BV12-3 though with variable CDR3 regions (I assume this is what the various grey pie segments means in figure 3). In both B*42 and B*81 donors, there are TCRs that recognize both B*81 and B42 bound to the peptide, and these also use BV12-3. In the latter case, there are large single clones – are these identical to the large single clones in the B81 responses in figure 3?

We apologize that Figure 3 was unclear. The reviewer is correct that each pie segment represents a unique TCR clone, and that all segments shaded in grey encoded BV12-3. We have clarified this point in the figure legend. Notably, despite very high usage of BV12-3, we observed no identical TCR beta sequences among the three B*81 participants. Furthermore, despite sharing BV12-3, none of the dual-reactive “public” TCR clones identified in B*42 participants were identical to any of the B*81-derived clones, indicating that TCR selection pressures may differ between these alleles.

The cross reactivity is puzzling in view of the different structures in the Kloverpris et al paper in Retrovirology. They show here the importance to T cell recognition of peptide residue 6 which is

orientated very differently in the two structures. It seems most likely therefore that the peptide flips into the B*81 orientation when the TCR binds, but that would need structural confirmation.

We agree with the reviewer's comment that this observation is puzzling. It is possible that interaction with TCR induces conformational changes in the peptide, resulting in an interaction surface that is more similar between B*81:01 and B*42:01. Alternatively, substitution of Asp at peptide position 6 may itself result in conformational changes that alter HLA binding or TCR recognition. A third possibility is that TCR encounter unique antigenic surfaces in the context of B*81:01 versus B*42:01, but Asp plays a critical role in both cases. Any of these models could help to explain the apparent selection of similar but non-identical TCR clonotypes. Unfortunately, we cannot address this question using our data and further structural studies will be needed. Due to the extensive effort required to obtain and analyse such complex structural datasets, we believe that this is beyond the scope of our current manuscript.

They go on to test cloned TCRs of the three types on cells pulsed with 180 variant peptides, covering all changes at each of the nine amino acids. It would be helpful if they could test each of these for binding to the two HLA molecules so that the non-binders can be discriminated from those that only interact with the TCR. Also they should indicate what peptide concentration was used; it would be better if these experiments were done on a minimum of three peptide concentration, as high concentrations are likely to minimize the differences and low concentrations would enhance them. That might be too difficult to do for all 180 variants but at least the peptide should be titrated for the wild-type sequence and they should show that they chose a suitable concentration to test the variant peptides. Similarly the peptide concentrations used in the experiment shown in figure 6 should be given and titrated.

We kindly ask the Reviewer to see our responses to Reviewer 1 regarding the peptide concentrations used in our studies (~20 μ M) and our results for TL9 peptide titrations. We agree that dilution studies will be needed to validate the ability of TCR to recognize variant peptides, particularly those that have not been examined previously; however, because this will require substantial time/effort and not significantly impact the conclusions of our study, we feel that it is beyond the scope of the current manuscript.

The Reviewer is correct that our reporter assay does not distinguish between TCR recognition and HLA binding. Since we focus our attention only on TL9 variants that are recognized by one or more TCR clones, inclusion or exclusion of non-binders is not expected to alter the conclusions of our study. Furthermore, while it may be preferable to exclude non-binders from our analyses, identifying such peptides is imprecise using current methods. Direct binding assays are labor-intensive and may not isolate low-affinity binders that are biologically relevant. Similarly, bioinformatics tools (e.g. netMHC; www.cbs.dtu.dk/services/NetMHC/) may be unreliable due to limitations of their training datasets. Prior work by Honeyborne et al (J Immunol 2006) and Kloverpris et al (J Virol 2012) described a strong preference of B*42:01 allele for Pro at position 2 and Leu, Met, Ile, or Phe at position 9, which are consistent with the TL9 sequence. Among B*42:01-derived TCR in our study, we observed robust recognition (>0.3 units) for two TL9 variants at position 2 (Ala and Pro) and six variants at position 9 (Thr, Val, Leu, Iso, Phe, and Met), suggesting a broader peptide binding motif than might be expected based on

direct binding assays. These seven variants are considered to be strong or weak binders according to the netMHC-4.0 algorithm:

Peptide	nM binding	%Rank	
TPQDLNTML	24.95	0.01	<= Strong Binder
TAQDLNTML	2432.51	1.80	<= Weak Binder
TPQDLNTMT	359.20	0.40	<= Strong Binder
TPQDLNTMV	95.06	0.12	<= Strong Binder
TPQDLNTMI	39.84	0.04	<= Strong Binder
TPQDLNTMP	925.43	0.90	<= Weak Binder
TPQDLNTMM	20.69	0.01	<= Strong Binder

A complete story would show the structures of the critical TCRs complexed to B*81 and B*42. That is lot to ask but it is the obvious question and it is possible that they are already doing this. If so, waiting to include these data would make the paper much stronger. Without this information it we are left a bit in the air as to how all this works. If the TCR flips the peptide conformation, I think that would be a ‘first’, although Tynan, Rossjohn et al have described a TCR flattening a bulging peptide (Nature Immunology 2007).

We agree with the reviewer that structural data would offer considerable mechanistic insight into how different TCR clones interact with TL9 in the context of B*81 versus B*42 - and possibility support the idea that TCR engagement “flips” the conformation of the peptide within the HLA binding groove (which would indeed be a novel observation). Solving tri-partite crystal structures of TCR/peptide/HLA complexes is not trivial and we are glad that the reviewer acknowledges that this may be too much to ask for the current manuscript. We are excited to explore this avenue of research using our TCR clones and hope to provide more detailed structural data in a future study.

The overall conclusion is that more broadly cross reactive T cells are better and control virus better, limiting escape. It is important that when patients are divided into those that do or do not they express cross reactive receptors, virus loads were significantly lower in the former, possibly better than the B*81 protection. The fact that both B81 and B42 can elicit such receptors, but that B42 does so less efficiently (presumably because of the different peptide orientation) is intriguing.

We thank both Reviewers for their constructive comments and suggestions.

Reviewers' comments:

Reviewer #1 (Remarks to the Author):

"Notably, intra-patient comparison of responses in either B*81:01 or B*42:01 participants showed that TL9 responses were more dominant compared to other responses ($p=0.03$) (Fig. S1)."

"aggregate data (B) show that TL9 is maintained at significantly higher frequencies than other responses."

This is somewhat better, but too vague. Were all the "other responses" tested by tetramer? Which responses were tested, and how were they selected, were epitopes selected by published epitopes?

I recommend including exactly which responses were tested in each individual (this could be done in table format for each individual, especially since it appears it was done at the epitope level), and how the total for "other responses" was derived.

"We examined TCR recognition using serial dilutions of consensus TL9 peptide (see Figure below) and now report these results as "data not shown" on lines 209-212 of the revised manuscript."

And later the authors refer to the serial dilutions again:

Line 348-340: We observed similar differences in the sensitivity of these clones over a range of TL9 concentrations (data not shown),

The data are informative and mentioned a couple times in the manuscript. It is also one of the few times a peptide titration was done. I recommend including this figure as supplementary data.

Reviewer #2 (Remarks to the Author):

The revised manuscript addresses the points raised by both reviewers satisfactorily. It is a pity that we are still rather uncertain about the exact explanation for the cross-reactivity which can only be answered by a structural analysis. However, as they do not have these data, I think we have to leave this point open at this time. There are many other interesting points made in this manuscript.

One small further point - I do think that they should show the titration figure for the TL9 peptides that they suggest should be 'data not shown'. This should be put in supplementary data.

Response to reviewers-Round 2

Reviewer #1 (Remarks for the Author):

“Notably, intra-patient comparison of responses in either B*81:01 or B*42:01 participants showed that TL9 responses were more dominant compared to other responses (p=0.03) (Fig. S1).”

“aggregate data (B) show that TL9 is maintained at significantly higher frequencies than other responses.”

This is somewhat better, but too vague. Were all the “other responses” tested by tetramer? Which responses were tested, and how were they selected, were epitopes selected by published epitopes?

I recommend including exactly which responses were tested in each individual (this could be done in table format for each individual, especially since it appears it was done at the epitope level), and how the total for “other responses” was derived.

We thank the reviewer for this suggestion. We have added a new table (Table 3) that summarizes the “other responses” tested in our study participants. These were tested by tetramer staining, as indicated in table 3 and lines 948-954, highlighted in yellow in the text. The list of tetramers was selected based on published epitopes and tetramer availability (also stated in table 3). The “other responses” in Figure S1B are plotted individually on the y-axis and the raw values can be found in table S1. Furthermore, we have included ELISPOT data from our study participants (Figure. S1C), these results also indicate TL9 is having the largest magnitude of response, further confirming TL9 as the most dominant epitope.

“We examined TCR recognition using serial dilutions of consensus TL9 peptide (see Figure below) and now report these results as “data not shown” on lines 209-212 of the revised manuscript.”

And later the authors refer to the serial dilutions again:

Line 348-340: We observed similar differences in the sensitivity of these clones over a range of TL9 concentrations (data not shown),

The data are informative and mentioned a couple times in the manuscript. It is also one of the few times a peptide titration was done. I recommend including this figure as supplementary data.

We thank the reviewer for this suggestion and have now included a supplementary figure (Figure S3) that shows the results of TL9 peptide titrations on B*42+ target cells for representative mono- and dual-reactive TCR clones. Furthermore, we have modified the text to reflect these changes, including a new figure legend on lines 967-975.

Reviewer #2 (Remarks to the Author):

The revised manuscript addresses the points raised by both reviewers satisfactorily. It is a pity that we are still rather uncertain about the exact explanation for the cross-reactivity which can only be answered by a structural analysis. However, as they do not have these data, I think we have to leave this point open at this time. There are many other interesting points made in this manuscript.

We thank the review for the feedback and agree with the reviewer that that in the absence of structural analysis, we are unable to define structural determinants of TCR cross-reactivity. To reconcile this we have now stated this as a limitation in the manuscript indicated in lines 389-392. However, we hope that our comprehensive analysis of peptide cross-recognition may contribute to potential future studies.

One small further point - I do think that they should show the titration figure for the TL9 peptides that they suggest should be 'data not shown'. This should be put in supplementary data.

We kindly ask the Reviewer to see our response to reviewer 1 as we have now included this data as supplementary figure.

We thank both reviewers for their constructive comments and suggestions.

REVIEWERS' COMMENTS:

Reviewer #1 (Remarks to the Author):

The authors have done a nice job incorporating reviewer comments into the revised manuscript. The additional supplemental figures/tables have improved the manuscript significantly.

Reviewer #2 (Remarks to the Author):

The authors have made further improvements to the manuscript that address my concerns.

Response to Reviewers:

Reviewer #1 (Remarks to the Author):

The authors have done a nice job incorporating reviewer comments into the revised manuscript. The additional supplemental figures/tables have improved the manuscript significantly.

We thank the reviewer for their positive and encouraging comments.

Reviewer #2 (Remarks to the Author):

The authors have made further improvements to the manuscript that address my concerns.

We thank the reviewer for their positive and encouraging comments.